# Inhibitory Effect of Isoliquiritigenin in Niemann-Pick C1-Like 1-Mediated Cholesterol Uptake

**DOI:** 10.3390/molecules27217494

**Published:** 2022-11-03

**Authors:** Jun Zeng, Wenjing Liu, Bing Liang, Lingyu Shi, Shanbo Yang, Jingsen Meng, Jing Chang, Xiaokun Hu, Renshuai Zhang, Dongming Xing

**Affiliations:** 1Cancer Institute, The Affiliated Hospital of Qingdao University, School of Basic Medicine of Qingdao University, Qingdao 266000, China; 2Qingdao Cancer Institute, Qingdao 266000, China; 3Department of Neurosurgery, The Affiliated Hospital of Qingdao University, Qingdao 266000, China; 4School of Life Sciences, Tsinghua University, Beijing 100010, China

**Keywords:** Niemann-Pick C1-Like 1 (NPC1L1), inhibitors, isoliquiritigenin, cholesterol uptake, cardiovascular disease

## Abstract

Isoliquiritigenin (ISL) is a flavonoid with a chalcone structure extracted from the natural herb *Glycyrrhiza glabra*. Its anti-inflammatory, antibacterial, antioxidant, and anticancer activities have been extensively studied. Moreover, ISL also possess hypolipidemic and atherosclerosis-reducing effects. However, its cholesterol-lowering mechanisms have not been reported yet. Niemann Pick C1 Like 1 (NPC1L1) is a specific transporter of cholesterol uptake. In this study, we found for the first time that ISL downregulates NPC1L1 expression and competitively inhibits cellular cholesterol uptake by binding to NPC1L1 in a concentration-dependent manner in vitro. This study provides a theoretical basis for further investigation of the molecular mechanisms of its cholesterol-lowering effect in vivo and inspired emerging drug research for cholesterol-lowering purposes through NPC1L1 inhibition.

## 1. Introduction

Cholesterol, as an essential component of cells, gives their membranes strength and flexibility. It is also associated with many metabolic pathways in the body, including the production of steroid vitamin D, bile, and hormones [1]. It is well known that hypercholesterolemia is a recognized cause of cardiovascular morbidity and mortality worldwide, and therefore cholesterol is considered one of the major factors in cardiovascular pathology [2]. Several other common comorbidities are associated with hypercholesterolemia, including diabetes, fatty liver disease, Alzheimer’s disease, gallstones, and some cancers. Elevated plasma cholesterol levels can lead to atherosclerosis and coronary artery disease [3], and patients often have hypertension, obesity, and hyperglycemia. Plasma cholesterol level is closely related to intestinal absorption, de novo biosynthesis, and cholesterol removal from blood [4]. Therefore, the inhibition of intestinal absorption, cholesterol biosynthesis, and the promotion of cholesterol excretion can all be therapeutic approaches to prevent and treat hypercholesterolemia. Cholesterol absorption is a complex multi-step process, in which cholesterol is firstly micellarized by bile acid in the intestinal cavity, then absorbed by intestinal cells, and then assembled into lipoproteins, and finally transported to the lymphatic and circulatory systems. Regulating cholesterol homeostasis is closely related to the intestine and can reduce plasma cholesterol by approximately 36% by completely inhibiting cholesterol absorption [5].

Niemann Pick C1 Like 1 (NPC1L1), a recently identified protein associated with cholesterol uptake, has been identified as a specific transporter of cholesterol uptake. It is highly expressed in the small intestine and liver [6]. Cholesterol uptake in the small intestine and liver was closely associated with NPC1L1 (Figure 1A) [7]. Studies have shown that NPC1L1 knockdown mice significantly reduced cholesterol absorption, which indicated that NPC1L has a vital role in promoting intestinal cholesterol absorption [8,9]. NPC1L1 facilitates cholesterol transport by lattice-protein-dependent endocytosis. NPC1L1 is first surrounded by lattice-protein-encased vesicles, then transported into the cytoplasm and ultimately into the recycled endosome. In the cutosol, cholesterol dissociated from NPLC1L1 and translocated to the endoplasmic reticulum. Meanwhile, NPC1L1 returned to the cytoplasmic membrane [10] (Figure 1B). When cholesterol uptake is reduced by inhibiting NPC1L1, the uptake of fat-soluble nutrients (such as triglycerides, bile acids, or fat-soluble vitamins) is not affected [11]. Therefore, NPC1L1 is a potential and promising target to lower blood cholesterol levels. According to our preliminary research on NPC1L1 inhibitors, very few NPC1L1 inhibitors have been reported. Currently, only ezetimibe (EZ) is approved by the FDA as an NPC1L1 inhibitor for clinical cholesterol-lowering therapy [12,13]. Therefore, the development of novel NPC1L1 inhibitors is vital for the treatment of diseases such as hypercholesterolemia.

Natural medicines are increasingly being used to prevent and treat human diseases. Not only all drugs used in traditional medicine worldwide of natural origin, but many more are produced by structural modification and modification of natural products as lead compounds [14,15,16]. Natural products, characterized by their structural and biological activity diversity, provide a direct route to the development of new drugs or drug lead compounds, as well as providing inspiration for chemical synthesis and structural. Optimizing natural products as a source of drug precursor molecules and their development process can facilitate drug development to benefit human health. Surveys showed that between 1981 and 2014, more than 59% of the new drugs approved by the US FDA, directly or indirectly, came from natural products [17]. Isoliquiritigenin (ISL) is extracted from the herb *Glycyrrhiza glabra* and is a flavonoid with a chalcone structure with the molecular formula C_15_H_12_O_4_ [18]. It has been reported that ISL has various pharmacological properties, not only antioxidant, antibacterial, anti-inflammatory, and anticancer activities, but also plays a role in liver protection, heart protection, and immune regulation [19,20,21,22]. Previous studies have shown that ISL inhibits adipose tissue inflammation and liver injury caused by a high-fat diet by effectively inhibiting the activation of the nucleotide-binding domain, the leucine-rich repeat sequence family, and the pyridine-containing domain of the 3-inflammasome [19]. ISL also inhibits hepatic steatosis by reducing fat accumulation and inhibiting adipogenic genes in mice fed a high-fat diet [23]. Furthermore, ISL inhibits lipid accumulation and insulin signaling by activating tyrosine phosphatase 1B [24]. In addition, ISL can improve plasma lipid levels and attenuates atherosclerosis in Apolipoprotein E-deficient mice [25]. Importantly, ISL has also been reported to have hypolipidemic and atherosclerotic effects [25,26,27,28]. However, the target and mechanism of its lipid-lowering (cholesterol-lowering) effects are unclear. In this study, we found that ISL can reduce cellular uptake of cholesterol by inhibiting NPC1L1. We initially explored its mechanism of action through BIAcore assay and molecular docking assay, which can provide a theoretical basis for the screening and development of novel NPC1L1 inhibitors.

## 2. Materials and Methods

### 2.1. Materials and Reagents

ISL, EZ (purity > 95%), and (R)-Mevalonic acid lithium salt were purchased from Sigma, New York, NY, USA. Niemann-Pick C1-like 1, β-actin, β-tubulin (antibodies were purchased from ABclonal, Wuhan, China), fetal bovine serum, Dulbecco’s modified Eagle’s medium (DMEM) and cell lysis buffer (10×), secondary antibody horseradish peroxidase-conjugated anti-rabbit (L3012), ultra-enhanced chemiluminescence detection reagent, Filipin bacteriocin, and BCA protein average/concentration detection kit were all purchased from Meilun Biotechnology (Dalian, China). Sodium taurocholate hydrate and lovastatin were purchased from Maclean’s, NBD-labeled cholesterol (from J&K scientific, Beijing, China).

### 2.2. Cell Culture 

Cells were cultured in DMEM containing high glucose with stable l-glutamine, 10% fetal bovine serum, 100 U/mL penicillin, and 100 μg/mL streptomycin. Cells were cultured in a humidified incubator at 37 °C, 95% air, and 5% CO_2_. The medium was changed every other day, and cells were passaged at 1:2 after reaching 70% to 80% cell confluency.

### 2.3. Cell Viability Assay 

Cell viability was determined by the CCK-8 cell proliferation assay using WST-8 cleavage. Briefly, 100 μL of HepG2 cells/Caco-2 cells (about 3000 cells per well) were inoculated into 96-well plates for 24 h. Then after treatment with/without 100 Μm ISL for 24 h, the cells in each well were reacted with 10% CCK-8 for 1 h. Optical density (OD) values were then detected at 450 nm using a microplate reader. 

### 2.4. Western Blot Analysis

#### 2.4.1. Extraction and Preparation of Total Cellular Proteins

The attached cells were digested with trypsin after being washed twice with PBS at 4 °C. Then, collected cell suspensions were put into EP tubes. Centrifuging with 1000 rpm, 5 min and removed supernatant, then resuspended with PBS, repeated centrifugation, and removed supernatant. The precipitate was added with 50–100 μL of cell lysate and lysed on ice for 30 min (or overnight at −20 °C). Then, centrifuge at 1000 rpm and 4 °C for 10 min. The supernatant was transferred to a new EP tube and set aside. The protein concentration in the cell lysate was determined using the BCA Protein Assay Kit (Pierce Biotechnology Inc., Rockford, IL, USA).

#### 2.4.2. Protein Blot Analysis

Equal amounts of protein (10–20 μg) were electrophoresed on SDS-PAGE gels (8%) and were then transferred to polyvinylidene fluoride (PVDF) membranes. Membranes were cut according to the position of the target and internal reference proteins and then incubated with a specific primary antibody (1:1000) overnight at 4 °C with anti-NPC1L1 antibody and anti-internal reference protein antibody (1:5000). PVDF membranes were washed three times with TBST followed by incubation with horseradish peroxidase-coupled secondary antibody (1:50,000) for 2 h. Finally, specific NPC1L1 bands (145 kD) and internal reference protein bands were identified with enhanced chemiluminescence advanced reagents.

### 2.5. Molecular Docking Methods

This experiment was conducted as described previously [29]. This experiment was conducted as previously described. Briefly, docking simulations were carried out utilizing the SYBYL-X 2.0 software. All the ligand molecules were drawn with the standard parameters of SYBYL-X. The Gasteiger-Huckel charges of protein receptors were prepared by standard methods using Tripos force field for 1000 steps to minimize the geometric conformation energy. The H-bonds were shown using a dotted line. Pymol was used to observe the interaction between the ligands and the protein receptors.

### 2.6. NBD-Cholesterol Uptake Assays

Some refinement of the previously reported method [30]. Cholesterol uptake by HepG2 cells and Caco-2 cells was assayed. Briefly, cells were seeded in 96-well plates (about 3000 cells/well). After culturing in a complete DMEM medium for 24 h, the cells were washed twice with PBS at 4 °C. Cells were cultured for 24 h by adding an equal amount of serum-free DMEM medium containing 1% penicillin/streptomycin and 1% non-essential amino acids. The samples were then washed twice with PBS at 4 °C before adding different concentrations of samples in solubilized DMSO (12.5 μM–100 μM; diluted in DMEM containing 0.5 mM taurine sodium). After co-incubation with drugs for 4 h, DMEM containing 0.5 mM taurine sodium and 25 μM NBD-cholesterol was added and incubated for another 1 h. After washing twice with PBS at 4 °C, the fluorescence values were measured at excitation wavelength 485 nm and emission wavelength 535 nm. EZ was used as a positive control in this study.

### 2.7. Filipin Bacteriocin Staining

Filipin (dissolved in DMSO to make 25 mg/mL) was diluted in PBS/10% FBS to a final concentration of 0.05 mg/mL and used as a working solution. Cells were cultured in confocal culture chambers at different drug concentrations. After being washed three times with PBS, drug-treated HepG2 cells were added to 4% formaldehyde and fixed at room temperature for 10 min. The staining was then incubated with Filipin for two hours. Then, cells were washed with PBS at 4 °C three times. Images were captured using a Nikon confocal microscope (Nikon Instruments, Tokyo, Japan).

### 2.8. Molecular Kinetic Analysis

Kinetic analysis was performed with NBD-cholesterol, as described above. Different concentrations of drugs (200, 100, 10 μM) were measured with different concentrations of NBD-cholesterol as substrate (50, 25, 6.25, 3.125, 1.5625, 0.78125 μM). Fluorescence values were detected at excitation wavelength 485 nm and emission wavelength 535 nm. Then, we made a Lineweaver-Burk double reciprocal chart to determine the type of inhibition.

### 2.9. SPR

SPR experiments were performed using Cytia T200 (CM5 chip was from Wancheng, Cytia, Newark, DE, USA). The NPC1L1 protein was coupled to the CM5 chip surface using the Amine Coupling Kit with the following procedure. First, the CM5 chip was installed and primed according to the standard instrument procedure, followed by mixing NHS and EDC in a 1:1 ratio and running the mixture with a total volume of 200 μL, a flow rate of 10 μL/min, and a time of 600 s. Then the NPC1L1 protein was diluted to 20 μg/mL in sodium acetate solution at pH = 4.6 and run with a total volume of 800 μL of the protein solution and a flow rate of 10 μL/min for 3000 s. Finally, the ethanolamine solution was run with a total volume of 200 μL and a flow rate of 10 μL/min for 420 s. Then, the CM5 chips were primed using a PBS-P solution containing 1% DMSO. ISL was diluted to different concentrations using PBS-P buffer containing DMSO and ensuring that the final concentration of DMSO was 1%. Different concentrations of ISL were run with a flow rate of 30 μL/min, a binding time of 60 s, and a dissociation time of 120 s. The final results were analyzed by the Biacore.T200.Evaluation software and Z software.

### 2.10. Statistical Analysis

All experiments were repeated 2–4 times, and at least three experimental samples were processed for use in each experiment. The statistical results for each replicated experiment were essentially the same. Experimental data were analyzed using the GraphPad Prism (v.6.01) software. Statistical significance between samples was determined by one-way ANOVA (Tukey’s test). *p*-values < 0.05 were considered statistically significant.

## 3. Results

### 3.1. SPR Results Show an Interaction between ISL and NPC1L1

Since NPC1L1 is a transporter protein with no catalytic activity, the inhibitory effect of ISL against NPC1L1 could not be evaluated by conventional colorimetric methods. Thus, the binding of ISL to NPC1L1 was analyzed by surface plasmon resonance (SPR). SPR is a method that can characterize the interaction between target proteins and molecules and has been widely used in drug discovery. According to the response value and SPR trace results in Figure 2, EZ interacts with NPC1L1, and the response value increases with increasing ISL concentration, i.e., the interaction with NPC1L1 is stronger, which is consistent with the previously reported results [29]. ISL also exhibits the same interaction as EZ; the response values increase with increasing concentration, and the interaction is stronger. Therefore, based on the results shown by SPR, the inhibitory effect of ISL can be further evaluated.

### 3.2. Results of Protein Immunoblot Analysis Defining the Cell Model

To screen appropriate cell lines for further experiments, we performed protein immunoassays to determine NPC1L1 levels in six cell types: cervical cancer cell line (Hela), colon cancer cell line (Caco-2), human osteosarcoma cells (U-2OS), human pancreatic cancer cells (SW1990), human breast cancer cell line (MCF-7), and liver cancer cell line (HepG2). The resultant bands were also analyzed for intergroup protein levels. The results showed high expression of NPC1L1 in HepG2 and Caco-2 and low expression in U-2OS (Figure 3). HepG2 and Caco-2 cells had the widest protein bands, and U2OS had the narrowest protein bands (Appendix A). Meanwhile, optical densitometry of the protein blots showed the same results. Previous studies have reported NPC1L1 was highly expressed in the small intestine and liver [6]. Our results were consistent with the previously reported results [31]. We then used HepG2 and Caco-2 cells as cellular models to explore the role of ISL on NPC1L1.

### 3.3. ISL Inhibited Uptake of Cholesterol in HepG2/Caco-2 Cells 

NPC1L1 activity was investigated in HepG2/Caco2 cells. Cholesterol uptake in the intestine is mediated by NPC1L1, whose function can be inhibited explicitly by EZ [8,12] in a concentration-dependent manner [32]. After pretreatment of cells with ISL for 24 h, we found that cholesterol uptake in HepG2/Caco-2 cells increased with a decrease in ISL concentration (Figure 4). EZ was used as a positive control drug during this period, and 100 μM of EZ reduced cholesterol uptake by approximately 55% in HepG2 cells and by approximately 70% in Caco-2 cells. 

### 3.4. Filipin Staining Results Showed That ISL Reduced Cellular Uptake of Cholesterol

To further determine the effect of ISL on the activity of NPC1L1, we used filipin bacteriocin staining and confocal to visualize the cholesterol uptake. Filipin is an antibiotic that specifically binds free cholesterol. Changes in cellular cholesterol content were observed by confocal microscopy at different times, thus verifying whether ISL has an inhibitory effect on cellular cholesterol uptake. At the same time, U-2OS, which hardly expresses NPC1L1, was used as a control group.

Referring to Liang Ge [10] and Andrew J et al. [33], cells were processed as shown in Figure 5A. Firstly, the cells were cultured in cholesterol-depleting medium for 60 min to deplete the cholesterol in the cells. The cholesterol-depleting medium formulation consisted of DMEM with 5% LPDS, 50 mM mevalonate, 10 mM compactin, and 1.5% CDX. The cells were then treated with cholesterol medium supplemented with the drug for another 60 min. Cholesterol-replenishing medium contained DMEM supplemented with 5% LPDS, 50 mM mevalonate, 10 mM compactin, and various concentrations of cholesterol/CDX. Cells were stained after fixation at different time points. 100 μM of ISL concentration was used, and 30 μM of EZ was used as a positive control. The results in Figure 5D show that EZ resulted in reduced cholesterol uptake by the cells compared to the blank control, consistent with previous studies reported [10]. Similarly, ISL also reduced cellular uptake of cholesterol. However, there was no such trend in the positive control cells, and there was no difference between the drug and control groups. Relative cholesterol content was also quantified by quantification of intracellular fluorescence (Figure 5B,C). The results were consistent with those indicated in Figure 5D.

### 3.5. ISL Is a Competitive Inhibitor of NPC1L1

Preliminary results suggest that ISL consistently affects NPC1L1, which prompted us to next investigate its kinetics of NPC1L1 inhibition. As shown in Figure 6B, Lineweaver-Burk double inverse plots show ISL is a noncompetitive inhibitor of NPC1L1.

### 3.6. Molecular Docking Revealed the Binding Pocket of NPC1L1 with ISL

The docking simulation showed that four residues of NPC1L1, including Ser52, Ser102, His124, and Thr128, were involved in the interaction with ISL. The docking simulation showed that four residues of NPC1L1, including Ser52, Ser102, His124, and Thr128, were involved in the interaction with ISL. His124 was a crucial residue in the cholesterol binding pocket [34] (Figure 7). Thus, molecular docking results suggested that the ISL may be an ideal NPC1L1 inhibitor.

### 3.7. ISL Reduced the Expression of NPC1L1 in HepG2 Cells

The previous experiments demonstrated that ISL does have an inhibitory effect on NPC1L1-mediated cholesterol uptake. Here, we further investigated whether ISL affected the level of NPC1L1 in HepG2 cells. As shown in Figure 8, protein immunoblotting was performed after pretreatment of cells with different concentrations of ISL (100, 10, 1 μM) for 24 h. EZ-treated cells were used as a positive control. The results showed a decrease in NPC1L1 protein levels in the cells after ISL action, which exhibited the same trend as the positive control. It indicates that ISL significantly reduced NPC1L1 protein levels in a dose-dependent manner (Appendix A).

### 3.8. ISL Has Low Cytotoxicity against HepG2/Caco-2 Cells

Finally, we tested whether ISL is cytotoxic. The cellular activity was measured using CCK8 assay, and the results showed that 100 μM of ISL had minimal effect on the cell survival of HepG2 and Caco-2 cells. As shown in Figure 9, the survival rate was about 87% for HepG2 cells and 93% for Caco-2 cells (Figure 9).

## 4. Discussions

NPC1L1 is a crucial transporter protein for cholesterol uptake [8]. It was not until 2004 that NPC1L1 was shown to be an essential transporter protein mediating intestinal cholesterol absorption. Subsequent studies confirmed that NPC1L1 is also the target of action of the new lipid-lowering drug EZ, which is widely used in clinical practice. Natural products have received more and more attention as a treatment for hypercholesterolemia. Previous studies have shown that some natural products have cholesterol-lowering effects through NPC1L1. For example, curcumin inhibited cholesterol uptake by reducing the protein and mRNA expression levels of NPC1L1 in intestinal Caco-2 cells [32]. Hawk tea extract induced transcription downstream of the LDL receptor, thereby inhibiting the uptake of free cholesterol by NPC1L1 [35]. Fomiroid A interfered with the action of glucosinolates with NPC1L1 and dose-dependently blocked NPC1L1-mediated cholesterol uptake and formation [36]. Lignans and quercetin inhibited NPC1L1, causing the decrease in intestinal cholesterol absorption [37]. ISL is a natural product derived from the natural herb *Glycyrrhiza glabra*. It has been reported that ISL has lipid-regulating and atherosclerosis-reducing functions. ISL blocked insulin-induced ROS production and inhibited lipid accumulation. ISL increased SR-BI expression in hepatocytes and thus promoted selective hepatic uptake of cholesterol to assist HDL catabolism. ISL controls obesity by regulating rate-limiting enzymes in the fatty acid synthesis and oxidation pathways in the liver. ISL also inhibited NF-κB and mitogen-activated protein kinase (MAPK) signaling pathways to slow the atherosclerotic process [24,25,26,27,28,38]. The present study showed that ISL significantly inhibited cholesterol uptake by HepG2 cells and Caco-2 cells. At a concentration of 100 μmol/L, the effect of ISL was comparable to that of the same dose of cholesterol absorption inhibitor EZ, a lipid-lowering drug, which was also visually demonstrated in the results of the Phillip staining assay in this study. When the cells were treated with drugs, immunoblotting results showed that the NPC1L1 protein was reduced by ISL. Thus, the results of this study reveal a potential new biological effect of ISL and its lipid-lowering mechanism, i.e., the lipid-lowering effect is achieved by inhibiting NPC1L1, thereby reducing the uptake of exogenous cholesterol.

In conclusion, the results of this study indicate that ISL is a competitive inhibitor of NPC1L1, which can significantly inhibit cellular uptake of cholesterol via NPC1L1 in a dose-dependent manner. Meanwhile, ISL has negligible cellular toxicity and thus can be further explored in vivo. The next further exploration of the molecular mechanism and in vivo effects of ISL in regulating NPC1L1 can provide a theoretical basis for the screening and development of novel NPC1L1 inhibitors. In addition, it is essential for the safe and effective regulation of lipids and the prevention of atherosclerosis, giving us a research basis for the prevention and treatment of cardiovascular diseases.

## Figures and Tables

**Figure 1 molecules-27-07494-f001:**
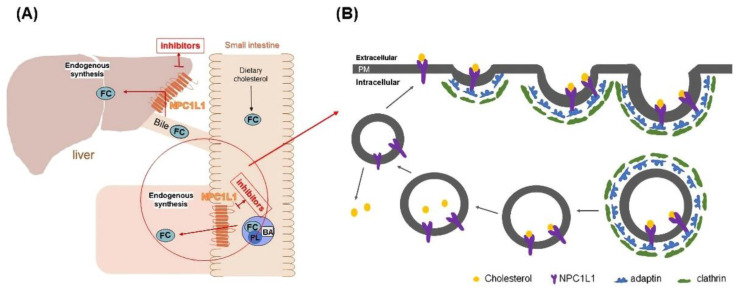
(**A**) Schematic diagram of the mechanism of NPC1L1 in small intestine and liver cholesterol transport. (**B**) Schematic diagram of NPC1L1-mediated cholesterol uptake process. (PL) phospholipids, (BA) bile acids, (FC) free cholesterol, and (PM) plasma membrane.

**Figure 2 molecules-27-07494-f002:**
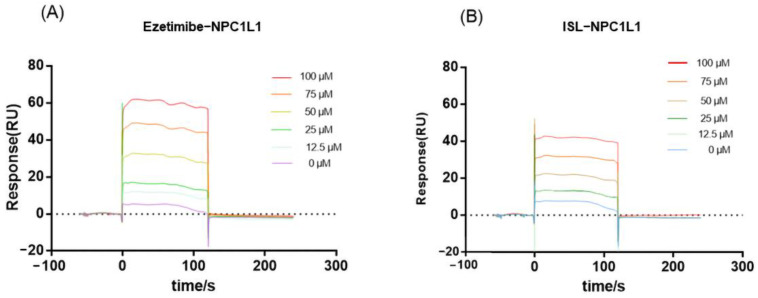
SPR assay of the interaction of ISL with NPC1L1. (**A**) Binding sensing map showing the interaction of EZ with immobilized NPC1L1. (**B**) Binding sensing map showing the interaction of ISL with immobilized NPC1L1. (Inhibitor concentrations were 0, 12.5, 25, 50, 75 and 100 μM).

**Figure 3 molecules-27-07494-f003:**
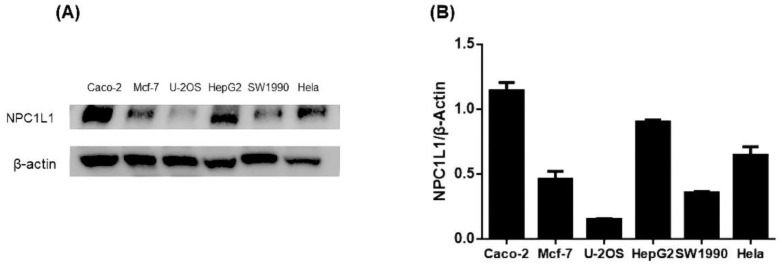
(**A**) NPC1L1 protein levels in HepG2, Caco-2, U-2OS, SW1990, MCF-7, and Hela with β-Actin as the internal reference protein. (**B**) Analysis of the optical densitometry of the protein blots.

**Figure 4 molecules-27-07494-f004:**
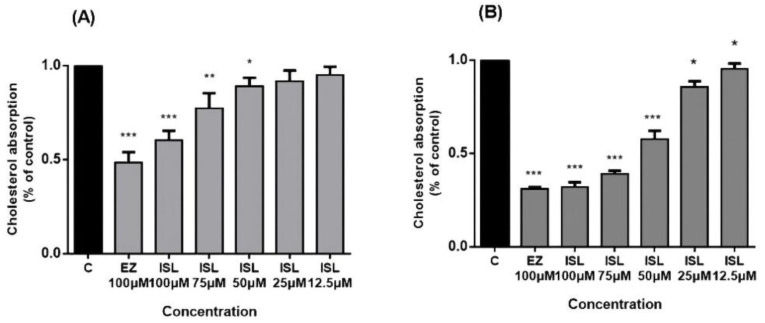
Inhibitory effect of ISL on cholesterol uptake in HepG2/Caco-2 cells. Cells were pretreated with different concentrations of ISL for 4 h and then incubated with radioactive micellar cholesterol for 1 h. In the absence of ISL, cholesterol uptake was normalized to 100%. Results are the mean ± SEM of three determinations in three independent experiments. (**A**) Inhibitory effect of ISL on cholesterol uptake in HepG2 cells. (**B**) Inhibitory effect of ISL on cholesterol uptake in Caco-2 cells. * *p* < 0.05, ** *p* < 0.001, *** *p* < 0.0001.

**Figure 5 molecules-27-07494-f005:**
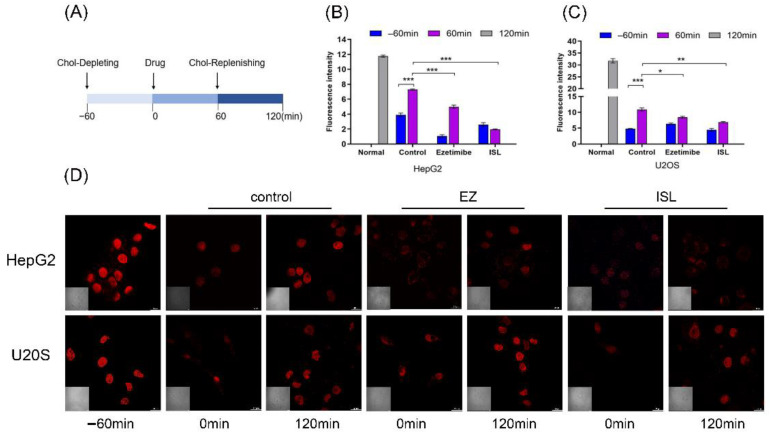
ISL inhibited cellular absorption of cholesterol by filipin bacteriocin staining. (**A**) The procedure used to treat the cells. (**B**,**C**) Quantification of total cholesterol of cells in (**D**). Error bars represent standard deviations. * *p* < 0.05, ** *p* < 0.001, *** *p* < 0.0001. (**D**) Cells were treated as shown in (**A**). Images were captured with Nikon confocal microscope after fixing cells and staining with filipin at different time points. Scale bar = 20 μm.

**Figure 6 molecules-27-07494-f006:**
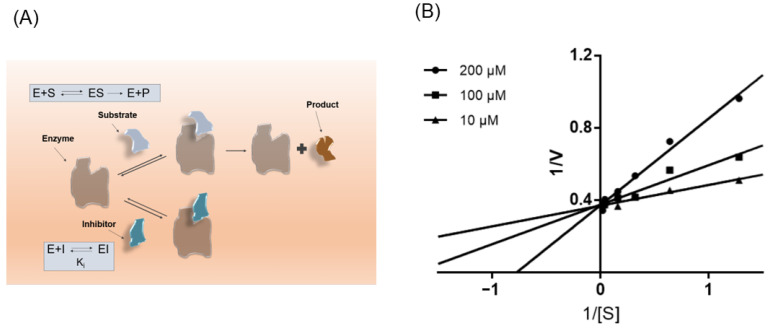
ISL is a noncompetitive inhibitor of NPC1L1. (**A**) Schematic diagram of the mode of action of competitive inhibitors. (**B**) Lineweaver-Burk plots in the presence of different concentrations of ISL.

**Figure 7 molecules-27-07494-f007:**
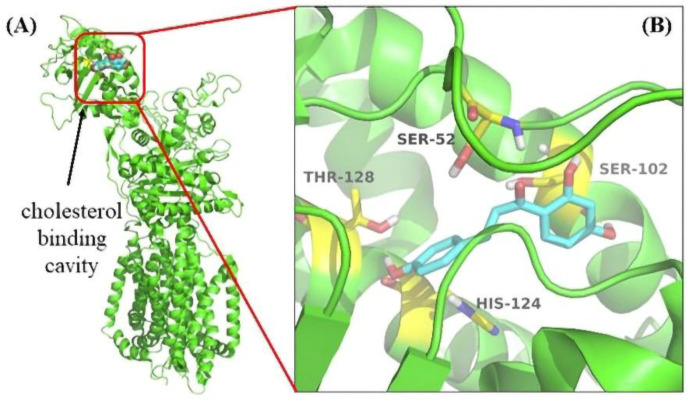
The result of Molecular Docking. (**A**) The cholesterol binding cavity (red) in the NTD. (**B**) Binding modes of ISL to NPC1L1 (PDB: 6V3F). The inhibitor ISL was shown color by element (carbon in cyan). The critical amino acid residues were shown color by element (carbon in yellow).

**Figure 8 molecules-27-07494-f008:**
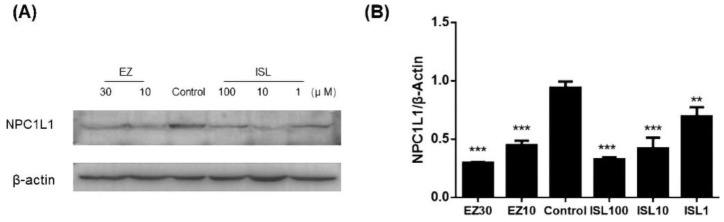
Effect of ISL on NPC1L1 protein expression in HepG2 cells. (**A**) Cells were treated with different concentrations of ISL for 24 h, and whole cell lysates were analyzed by protein blotting. (**B**) Analysis of the optical densitometry of the protein blots. ** *p* < 0.001, *** *p* < 0.0001.

**Figure 9 molecules-27-07494-f009:**
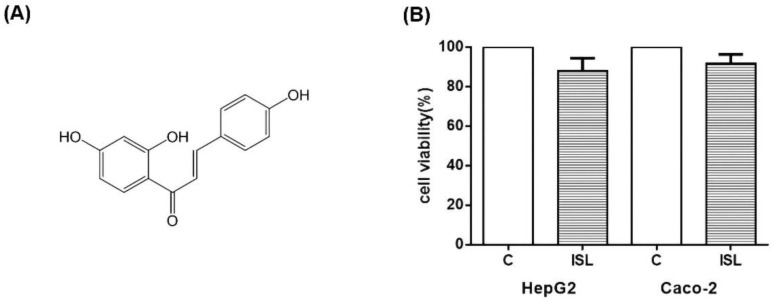
(**A**) Diagram of ISL molecular structure. (**B**) Effect of 100 μM ISL on cell viability. 100 μM ISL had a minimal effect on cell viability. The viability of HepG2 cells was approximately 87%, and that of Caco-2 cells was approximately 93%. Experiments were performed in triplicate and expressed as mean ± standard deviation.

## Data Availability

Not applicable.

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
