# Peer review of "Inhibitory Effect of Isoliquiritigenin in Niemann-Pick C1-Like 1-Mediated Cholesterol Uptake"

_molecules, 2022, doi:10.3390/molecules27217494_

Round 1
Reviewer 1 Report
The manuscript, entitled "Inhibitory Effect of Isoliquiritigenin in Niemann-Pick C1-like 1-mediated Cholesterol Uptake" focuses on new drug level studies for cholesterol detection by NPC1L1 inhibition.
The authors present interesting theoretical and experimental results.
However, there are a few notes:
1. Please, to the description of figure 1A, add a link to 6 sources from the list of references, since a similar figure is given in this article.
2. Should we add discussions of already existing traditional traditional medicines as a comparison with the data presented?
Author Response
Response to Reviewer 1 Comments
- Please, to the description of figure 1A, add a link to 6 sources from the list of references since a similar figure is given in this article.
Response:Thank you very much for pointing out this problem. As figure 1A was made by ourselves, thus links were not included. In the revised manuscript, we add the relevant references and highlighted them in red (page 2, line 47).
- Should we add discussions of already existing traditional medicines as a comparison with the data presented?
Response:Thanks for your suggestion. On page number 9, lines 306-315, we have discussed some natural products that have been reported to have inhibitory effects on NPC1L1, such as curcumin, fomiroid A, lignans, and quercetin. They all inhibited cholesterol uptake by affecting NPC1L1.
Reviewer 2 Report
1. page number 2, line -63.
Statement: More and more natural medicines are used to prevent and treat human diseases.
Suggestion: Double more does not make any sense
2. page number 2, line- 64.
Statement: Only are all drugs used
Suggestion: are not required
3. page number 2, line- 65.
Statement: Not only are all drugs used in traditional medicine worldwide of natural origin, but many more are produced by structural modification and modification of natural products as lead compounds.
Question: Any reference to prove that product produced by structural modification?
4. Material Methods: 2.3 cell viability assay, Line- 109-114
Statement: Then after treatment with/without 100 μM ISL for 24 112 h, the cells in each well were reacted with 10% CCK-8 for 1 h. Optical density (OD) values 113 were then detected at 450 nm using a microplate reader.
Suggestion: Authors requested to include different concentration of ISL in this part, it would be easy for future researcher.
5. Material Methods: 2.6 .NBD-Cholesterol uptake assays, Line -143-155
Question: on what basis the authors have fixed the concentration of NBD?
Concentration is higher than the previous research(0.1, 1, 5 and 10 µmol/l ( Authors can refer the article: https://doi.org/10.3892/mmr.2015.4154))
6. 3.2. SPR results show an interaction between ISL and NPC1L1. Line-179-190
Authors have done SPR of EZ and NPC1L1
Question: What are the consecutive injections at a flow rate? And authors could explain the detailed standard procedure in the material and method section.
7. 3.2.Results of protein immunoblot analysis efining the cell model: Line -195:
Question: What is efining?
8. 3.2. Results of protein immunoblot analysis efining the cell model, Line- 201
The results 200 showed high expression of NPC1L1 in HepG2 and Caco-2 and low expression in U-2OS
Suggestion: I suggest to include more details regarding the reason behind high and low expression of NPC1L1.
9. 3.6. Molecular Docking revealed the binding pocket of NPC1L1 with ISL. Line- 256-262
Question: In molecular docking, What was the Binding energy ΔG authors observed from the interaction
Author Response
Response to Reviewer 2 Comments
- page number 2, line -63.
Statement: More and more natural medicines are used to prevent and treat human diseases.
Suggestion: Double more does not make any sense
Response:Thanks for the comment. We modified this sentence to: Natural medicines are increasingly being used to prevent and treat human diseases. (line 63, page 2)
- page number 2, line- 64.
Statement: Only are all drugs used
Suggestion: are not required
Response: Sorry for the mistake. We checked the full text to ensure there were no other similar errors.
- page number 2, line- 65.
Statement: Not only are all drugs used in traditional medicine worldwide of natural origin but many more are produced by structural modification and modification of natural products as lead compounds.
Question: Any reference to prove that product is produced by structural modification?
Response:Thank you for your reminder, we have added the appropriate references[14-16]. (line 65, page 3)
- Material Methods: 2.3 cell viability assay, Line- 109-114
Statement: Then after treatment with/without 100 μM ISL for 24 112 h, the cells in each well were reacted with 10% CCK-8 for 1 h. Optical density (OD) values 113 were then detected at 450 nm using a microplate reader.
Suggestion: Authors requested to include different concentration of ISL in this part, it would be easy for the future researchers.
Response: Thanks for your comments. However, we only measured the cell viability of 100 μM ISL in this experiment. The 100μM concentration was large enough for ISL, so we did not do cell viability assay at other different concentrations.
- Material Methods: 2.6 .NBD-Cholesterol uptake assays, Line -143-155
Question: on what basis have the authors fixed the concentration of NBD?
Concentration is higher than the previous research (0.1, 1, 5 and 10 µmol/l ( Authors can refer to the article: https://doi.org/10.3892/mmr.2015.4154))
Response:Thanks for your comment. The concentration of NBD in our study was determined based to the previous study (doi:10.1186/s12906-019-2664-8;10.1091/mbc.E16-03-0154; 10.1016/j.heliyon.2020.e05408). We have chosen this concentration higher than the one used in your suggested literature (https://doi.org/10.3892/mmr.2015.4154), where the reason may be due to the use of a different cell model.
- 3.2. SPR results show an interaction between ISL and NPC1L1. Line-179-190
Authors have done SPR of EZ and NPC1L1
Question: What are the consecutive injections at a flow rate? And authors could explain the detailed standard procedure in the material and method section.
Response:Thank you for your suggestion. We were run at a flow rate of 30μL/min, the binding time was 60s, and the dissociation time was 120s. Moreover, we have added detailed procedures in the material and method section. We have marked the relevant content in red. (on page number 4, lines 170-184)
- 3.2. Results of protein immunoblot analysis efining the cell model: Line -195:
Question: What is efining?
Response:Sorry for the mistake. This is the spelling error, and we have corrected it to defining. (line 208, page 5)
- 3.2. Results of protein immunoblot analysis efining the cell model, Line- 201
The results 200 showed high expression of NPC1L1 in HepG2 and Caco-2 and low expression in U-2OS
Suggestion: I suggest to include more details regarding the reason behind high and low expression of NPC1L1.
Response: Thanks for your suggestion. We have added more details in this part. (line 2215-218, page 6)
HepG2 and Caco-2 cells had the widest protein bands, and U2OS had the narrowest protein bands. Meanwhile, optical densitometry of the protein blots showed the same results. Previous studies have reported that NPC1L1 was highly expressed in the small intestine and liver, which is consistent with our observations.
- 3.6. Molecular Docking revealed the binding pocket of NPC1L1 with ISL. Line- 256-262
Question: In molecular docking, What was the Binding energy ΔG authors observed from the interaction.
Response:Our docking simulations were carried out utilizing the SYBYL-X 2.0 software. We focused mainly on the interaction between our small molecules and the amino acids of NPC1L1. Therefore, there was no description of the binding energy ΔG.